# Factors Influencing the Difficulty and Need for External Help during Laparoscopic Appendectomy: Analysis of 485 Procedures from the Resident-1 Multicentre Trial

**DOI:** 10.3390/jpm12111904

**Published:** 2022-11-15

**Authors:** Stefano Piero Bernardo Cioffi, Andrea Spota, Michele Altomare, Stefano Granieri, Roberto Bini, Francesco Virdis, Federica Renzi, Elisa Reitano, Osvaldo Chiara, Stefania Cimbanassi

**Affiliations:** 1General Surgery and Trauma Team, ASST Niguarda, Milano, Piazza Ospedale Maggiore 3, 20162 Milan, Italy; 2Department of Surgical Sciences, Sapienza University of Rome, Piazzale Aldo Moro 5, 00185 Rome, Italy; 3Ospedale di Vimercate, Via Santi Cosma e Damiano 10, 20871 Vimercate, Italy; 4IRCAD Research Institute Against Digestive Cancer, 67091 Strasbourg, France; 5Department of Pathophysiology and Transplants, State University of Milan, Via Festa del Perdono 7, 20122 Milan, Italy; 6General Surgery Residency Program, University of Milan, Via Festa del Perdono 7, 20122 Milan, Italy

**Keywords:** laparoscopic appendectomy, technical difficulty, predictive factors, complicated appendicitis, laparoscopic surgery, obesity

## Abstract

Purpose: To identify preoperative predictive factors for technically challenging laparoscopic appendectomy (LA) and the need for external help to laparoscopically complete the procedure. Methods: We analysed data from a two-year data lock on the Resident-1 multicentre registry. The operator classified each procedure following a five-grade Likert scale to define technical difficulty. We performed univariate analysis comparing Grade 1–3 versus 4–5 procedures and then built a logistic regression model to identify independent predictors of Grade 4–5 procedures defined as needing external help to complete a LA. Results: 561 patients were recruited from 2019 to 2021, and 485 patients were included in the final analysis due to missing data. A BMI > 30 kg/m^2^, preoperative CT scan, and the AIR score were independent preoperative predictors of complex LA with the need for external help to be completed. Patients undergoing such procedures were more affected by CA, had longer operative times, and had the worst postoperative outcomes. Conclusion: The preoperative identification of technically demanding LA could be helpful in optimising the preoperative planning, maximise surgeons’ preparedness, and include expert surgeons in the procedure earlier. Creating a scoring system for the technical difficulty of LA is desirable.

## 1. Introduction

Acute appendicitis (AA) is globally one of the most frequent abdominal surgical emergencies, with a lifetime prevalence of 7–8% [1].

Laparoscopic appendectomy is one of the most performed procedures and is the standard of care for AA [2]. 

Complicated appendicitis (CA), defined as acute appendicitis with associated peritonitis, perforation, or intra-abdominal abscess (IAA), accounts for 14–55% of all cases of appendicitis. 

Laparoscopic appendectomy is a laparoscopic index procedure often performed by surgical residents, but in the case of CA, it can also be a true nightmare for experienced surgeons [2]. 

Some studies in the literature addressed the issue of technical difficulties during LA, exploring the learning curve of the procedure and comparing the outcomes of LA performed by residents and surgeons. 

Kim and colleagues studied the outcomes and learning curve of LA performed by residents under supervision. The authors defined the concept of surgical failure as the need for the supervisor to step in and complete the LA due to the high technical difficulty of the procedure. The surgical failure occurred in the case of massive adhesions, severe inflammation, or the identification of an abscess on preoperative imaging [3,4,5]. 

Two papers from Lin and Ussia showed how the type of appendicitis was significantly associated with operative time [6,7]. Ussia and colleagues also reported that the learning curve did not reach the expected plateau for high grades of appendicitis, considering operative time to be a quality indicator [7]. 

Other studies explored different surgical techniques, aiming to create specific scores predicting the technical difficulty of laparoscopic splenectomy for nontraumatic splenic disease, laparoscopic liver surgery, or laparoscopic spleen-preserving splenic hilar lymph node dissection for gastric cancer [8,9,10]^.^

Identifying factors associated with complex LA is essential for preoperative planning, improving mental preparedness, and optimising resource allocation.

This study aims to identify preoperative predictors of technically complex LA, defined as the need for an external operator to step in, scrubbed or not, to complete the procedure laparoscopically. 

## 2. Materials and Methods

This primary outcome of the study is the intraoperative need for external help from a more expert colleague to complete the procedure laparoscopically. 

The secondary outcomes are the intraoperative features and postoperative outcomes associated with technically complex LA.

REsiDENT-1 is a prospective resident-led multicentre observational trial that started in October 2019. The project aims to explore the relationships between PL and postoperative intraabdominal abscesses, introducing a classification for AA to standardise intraoperative grading, focusing on the appendix aspect and peritoneal contamination [11]. Data were extracted from a two-year data lock on the REsiDENT-1 trial multicentre registry.

The general-surgery residents of the University of Milan General Surgery residency program collected the data as planned in the REsiDENT-1 protocol [11].

Data entry started in October 2019, and the first data lock was performed in October 2021. The follow-up schedule was set at 90 days.

We considered the following criteria for patient enrolment.

Inclusion criteria:○Patients younger than 18 years old.○Surgical laparoscopic approach for AA.○Intraoperative and histological diagnosis of AA.Exclusion criteria:○Patients < 18 years old or > 80 years old.○Previous appendicitis treated conservatively.○Negative appendectomy.○Open approach for surgery or intraoperative conversion.○Coexistence of other intraabdominal infections (IAI).○Patients with immunodeficiency.○Patients treated with steroids, immunosuppressants, or CHT within the six previous months.

We collected pre-, intra-, and postoperative demographic data. Each AA was classified following our previously published classification proposal [11]. 

We applied a five-point Likert scale (Ls) [12] to assess LA difficulty and define the need for external support to laparoscopically complete the appendectomy.

A procedure performed by the first operator without any help from the assistant.The first surgeon can perform the procedure by seeking passive support from the assistant.The first surgeon can perform the procedure by seeking active support from the assistant.The procedure is technically demanding. The surgeon and the assistant can finish the procedure laparoscopically with the external support of a more expert nonscrubbed surgeon.The procedure is technically demanding and challenging. The surgeon and the assistant need the help of an external and more expert surgeon who scrubs in to safely finish the procedure laparoscopically.

Data were collected online using Google Forms (Google LLC, 1600 Amphitheatre Pkwy, 94043 Mountain View, CA) during the first year; data collection was then shifted to a Redcap online registry [13].

### Statistical Analysis

An independent-sample Student’s t or Mann–Whitney test was used for continuous variables. Continuous variables are expressed as mean with standard deviation or interquartile range according to distribution. Categorical variables are expressed as numbers and percentages. The distribution of each variable was studied with the Kolmogorov–Smirnov and Shapiro–Wilk tests. Differences in proportions were analysed with the Pearson χ^2^ or exact Fisher test. 

We performed univariate analysis to assess the relationship between variables and the need for external help to complete the LA, defined as difficulty Grade 4 or 5. 

Multivariate logistic regression was performed to identify prognostic factors for needing external help during laparoscopic appendectomy to complete the procedure. The variables to be included in the analysis were chosen following the results of univariate analysis, clinical evidence, and literature data. Data were checked for multicollinearity with the Belsley–Kuh–Welsch technique. The White and Shapiro–Wilk tests assessed the heteroskedasticity and normality of residuals. A *p*-value < 0.05 was considered to be statistically significant. Patients with missing data were excluded from analysis. Statistical analysis was performed with two different software programmes: online application EasyMedStat (version 3.17; www.easymedstat.com (accessed on 15 July 2022) and computer-based software IBM SPSS Statistics per Windows, Version 25.0 (IBM Corp., Armonk, NY, USA: IBM Corp.) 

## 3. Results

In total, 561 patients were recruited in the Resident-1 registry from October 2019 to October 2021 by 51 different surgical residents (see Appendix A for the complete list) from 21 hospitals within the network of the general-surgery residency programme of the University of Milan (see Appendix A for complete list). 

Group comparison analysis is reported in Table 1a for preoperative factors, Table 1b for intraoperative factors, and Table 1c for postoperative ones.

In total, 485 procedures were included in the logistic regression analysis due to missing data. The flow diagram of patient enrolment is reported in Figure 1. 

In multivariate analysis, the AIR score [14] (OR = 1.22, [1.06; 1.4], *p* = 0.0043), preoperative CT scan (OR = 2.38, [1.24; 4.57], *p* = 0.0092), and BMI > 30 kg/m^2^ (OR = 2.61, [1.29; 5.27], *p* = 0.0075) were associated with higher odds for the need for external help. The results of the logistic regression model are reported in Table 2. 

## 4. Discussion

In this spin-off of the Resident-1 prospective multicentre trial, we tried to identify preoperative factors independently associated with LA technical difficulty and needing external help to laparoscopically complete the procedure. We sought to identify a cluster of patients in which specific preoperative *warning signs* could help in resource allocation and early consultation by more expert surgeons to maximise surgical and clinical outcomes of patients with AA. 

Our main findings show that obesity, the need for a preoperative CT scan, and a progressively higher AIR score are risk factors for the need to call for help during LA.

In this cohort, AIR score was an independent preoperative risk factor for technical difficulty during LA and needing external help during surgery. Looking at the report of our univariate analysis in Table 1, patients who had undergone a technically demanding procedure also had significantly higher rates of CA with more elevated subsequent rates of postoperative complications. 

The AIR score was recently validated on a large multicentre cohort considering CA. The authors showed that the score best performed in discriminating CA compared to any appendicitis. An AIR score lower than 4 had a 99% negative predictive value for CA [14]. 

In our series, per each point of the AIR score, the odds for technical difficulty increased by 1.22 points. 

Our findings were confirmed in another paper published in 2018 in which the Alvarado and AIR scores were both associated with appendicitis severity considering the pathologist report. Only the AIR score was significantly related to the presence of a CA [15]. 

Looking at the composition of the AIR score comprising clinical and laboratory variables, it is easy to understand its value as an expression of appendicitis severity and subsequent technical difficulty of the LA. In fact, in our cohort, patients undergoing complex and challenging LA more frequently had a CA. 

Our results highlight how performing a preoperative CT scan more than doubles the odds for a challenging procedure with the need for external help to complete a LA. 

The most cited guidelines recommend performing a CT scan in the diagnostic pathway of AA only in the case of an ultrasound suggestive of CA in patients older than 40. Furthermore, a debated statement of the guidelines regarded patients under 40 years old with a higher risk score for AA, AIR 9–12—Alvarado 9–10—AAS > 15, in which cross-sectional imaging could be avoided by skipping to a diagnostic–therapeutic laparoscopy [16]. 

A recent multicentre retrospective study on 19,749 patients described the trend of diagnosing and managing patients affected by AA. An increasing trend towards using CT scans for diagnostic purposes was described regardless of appendicitis severity [17]. 

As also confirmed in the Jerusalem guidelines, other reports showed that a CT scan has good discrimination ability for CA to support the decision-making process towards nonoperative management or surgery [15]. A recent systematic review with meta-analysis showed a higher specificity but low sensitivity to identify CA during the diagnostic work-up with the CT scan [18].

Mahankali et al. progressed and published a CT-based predictive score to identify CA. A similar paper, published on the BJS in 2015, proposed a composite score based on clinical and CT scan features. which achieved good performance in discriminating uncomplicated vs. CA [19,20]. 

Looking at our data, performing a preoperative CT scan was an independent risk factor for a more complex procedure with the need for external support. More patients in this cluster underwent CT scans. Looking at the features of this subgroup, patients undergoing more complex procedures were older, 46.47 (±18.25) vs. 34.86 (±16.15) years old, with higher AIR and Alvarado scores [21], CCI [22], and higher rates of CA. The recommendations of the most recent guidelines were followed in our multicentre cohort despite the high heterogeneity due to the involvement of 21 different hospitals. 

Several studies explored the impact of obesity on the technical difficulty of laparoscopic surgery. In our research, operating on a patient with a BMI higher than 30 kg/m^2^ doubled the risk for a technically challenging LA with the need for external help. A meta-analysis from Qiu and colleagues explored the impact of obesity on surgical and postoperative outcomes of patients operated on for rectal cancer; the authors reported that obese patients had higher conversion rates and overall morbidity in terms of anastomotic leakage, wound infection, and pulmonary events [23].

These findings were partially confirmed by another study published on surgical endoscopy. The authors compared the short-term surgical outcomes of morbidly obese, obese, and nonobese patients undergoing laparoscopic colorectal surgery. They showed no differences in conversion rates, length of stay, anastomotic leakage, and 30-day readmission rates. Obesity only significantly impacted operative time: per each increasing point of BMI, operative time increased by roughly 2 min [24]. 

Another paper, by Li et al., focused on laparoscopic spleen-preserving splenic hilar lymph0node dissection for gastric cancer. It confirmed that obesity is an independent risk factor for the technical difficulty of the procedure. The authors created and validated a scoring system to evaluate the technical difficulty of the procedure [10]. 

Three more studies confirmed that visceral obesity was better correlated with outcomes for laparoscopic colon surgery compared to BMI [25,26,27]. 

Our findings are substantially confirmed by the available literature on laparoscopic surgery, although we did not find any reference on the specific relationship between technical challenges during LA and obesity. Looking at the result of our univariate analysis, we also confirmed some of the evidence reported above. In our series, a technically demanding procedure was associated with longer operative times, higher postoperative morbidity, especially in intrabdominal infections, and higher death rates. Considering 60-day outcomes, we did not report higher rates of new hospitalisations. Still, among those admitted for postoperative complications, we showed a significantly longer stay for the new episode in the case of complex procedures during the culprit hospitalisation. 

Dealing with complex LA can be mentally and technically demanding, often pushing surgeons outside their comfort zone. We already discussed how patients undergoing such procedures are more affected by CA. Considering our results, we also discovered how facing complex cases lead the surgeon to non-evidence-based choices. Patients undergoing complex LA needing external help underwent more frequent peritoneal lavage and intrabdominal drainage placement. Sometimes, the occurrence of intraoperative complications or dealing with a complex case can affect surgeon satisfaction and mental comfort, as shown by Erastam et al. This could be one of the reasons that, in our trial, pushed surgeons to adopting intraoperative solutions that were more based on habits rather than solid evidence [28].

We report the potential limitations and pitfalls in this spin-off related to the nature and design of the Resident-1 ongoing prospective trial. This is the first Italian resident-led prospective multicentre trial. To control the negative impact of limited experience, we spent a significant amount of time during the resident recruiting phase for education regarding data collection and follow-up. Despite this, we were unable to include all the patients in the logistic regression model and in the univariate analysis of postoperative outcomes due to missing data and patients lost to follow-up.

## 5. Conclusions

We demonstrated how specific preoperative features could suggest complex and challenging LA in which the external support of a more expert surgeon is needed to laparoscopically complete the procedure. These procedures are also more commonly performed in patients affected by CA with the worst postoperative outcomes, mainly related to the severity of appendicitis. 

A step forward should be elaborating a scoring system to predict the technical complexity of LA, which also needs to be validated on a larger cohort of patients. Our results and this future tool would be helpful in optimising resource allocation and planning before surgery, both considering the mental preparedness of the operator and making seniors surgeons on call be aware of the potential need for support during the LA.

## Figures and Tables

**Figure 1 jpm-12-01904-f001:**
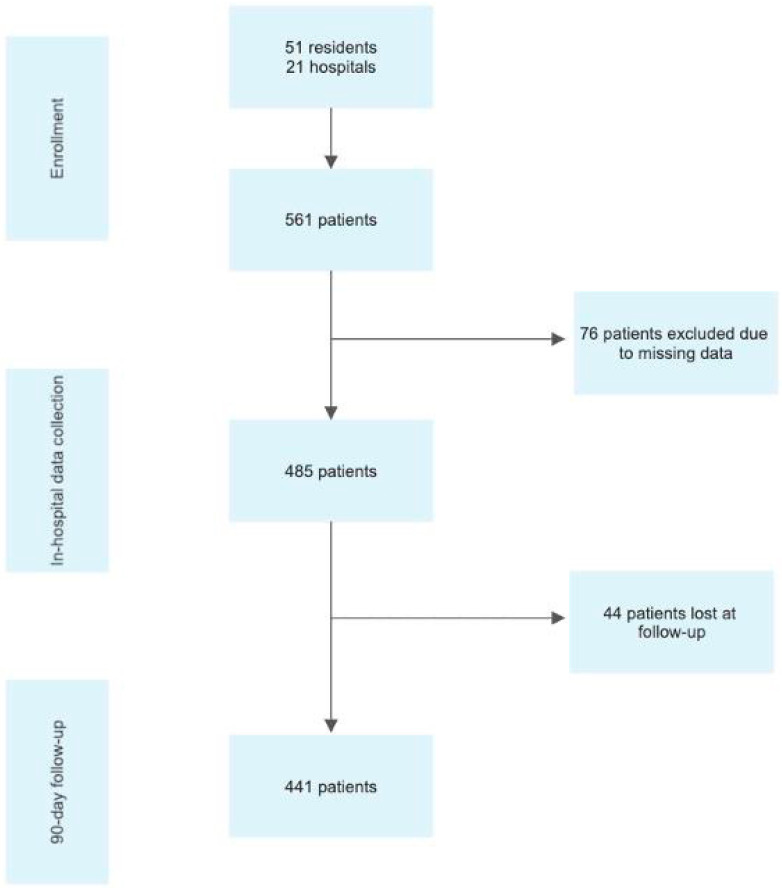
Flowchart of patient selection.

**Table 1 jpm-12-01904-t001:** (**a**) Comparison analysis considering preoperative variables between procedures with DG 4–5 and without needing external help (DG 1–3). DG, difficulty grade; BMI, body mass index; ASA, American Society of Anaesthesiology classification; CCI, Charlson comorbidity index; PIRO score, predisposition insult response organ dysfunction score; AIR score, appendicitis inflammatory response score; CT, computed tomography; US, ultrasound. (**b**) Comparison analysis considering intraoperative variables between procedures with DG 4–5 and without needing external help (DG 1–3). (**c**) Comparison analysis considering postoperative variables between procedures with DG 4–5 and without needing external help (DG 1–3). SSI, surgical site infection.

(a)			
Variable	DG 1–3	DG 4–5	*p*-Value
	N = 474	N = 87	
Age	34.86 (±16.15)	46.47 (±18.25)	<0.001
	95% CI: [33.4; 36.31]	95% CI: [42.58; 50.36]	
	Range: (19.0; 91.0)	Range: (18; 87.0)	
	N = 474	N = 87	
Sex			0.607
Female	197 (41.56%)	33 (37.93%)	
Male	277 (58.44%)	54 (62.07%)	
	N = 474	N = 87	
BMI > 30 kg/m^2^			<0.001
Yes	32 (6.75%)	21 (24.14%)	
No	442 (93.25%)	66 (75.86%)	
	N = 474	N = 87	
ASA			<0.001
1	293 (61.81%)	28 (32.18%)	
2	159 (33.54%)	48 (55.17%)	
3	21 (4.43%)	10 (11.49%)	
4	1 (0.21%)	1 (1.15%)	
	N = 474	N = 87	
CCI	0.403 (±1.02)	1.14 (±1.59)	<0.001
	95% CI: [0.311; 0.495]	95% CI: [0.8; 1.48]	
	Range: (0.0; 11.0)	Range: (0.0; 7.0)	
	N = 474	N = 87	
PIRO score	0.173 (±0.599)	0.632 (±1.26)	<0.001
	95% CI: [0.119; 0.227]	95% CI: [0.364; 0.9]	
	Range: (0.0; 8.0)	Range: (0.0; 9.0)	
	N = 474	N = 87	
Alvarado score	6.38 (±1.73)	6.99 (±1.42)	0.002
	95% CI: [6.22; 6.53]	95% CI: [6.69; 7.29]	
	Range: (1.0; 10.0)	Range: (3.0; 10.0)	
	N = 474	N = 87	
AIR score	5.73 (±1.94)	6.68 (±1.9)	<0.001
	95% CI: [5.55; 5.91]	95% CI: [6.27; 7.09]	
	Range: (1.0; 11.0)	Range: (2.0; 12.0)	
	N = 444	N = 85	
Preoperative exams			<0.001
CT	174 (39.82%)	59 (74.68%)	
US	263 (60.18%)	20 (25.32%)	
	N = 437	N = 79	
(**b**)			
**Variable**	**DG 1–3**	**DG 4–5**	** *p* ** **-Value**
	**N = 474**	**N = 87**	
Operative time	62.17 (±21.31)	92.94 (±33.91)	<0.001
	95% CI: [60.25; 64.1]	95% CI: [85.72; 100.17]	
	Range: (15.0; 180.0)	Range: (40.0; 230.0)	
	N = 474	N = 87	
Operator			0.009
Surgeon	311 (65.61%)	70 (80.46%)	
Resident	163 (34.39%)	17 (19.54%)	
	N = 474	N = 87	
Peritoneal contamination			<0.001
No contamination	273 (57.6%)	8 (9.2%)	
Single abscess	77 (16.24%)	25 (28.74%)	
Multiple abscess	4 (0.8%)	2 (2.3%)	
Localised purulent peritonitis	95 (20 %)	33 (37.9%)	
Diffuse purulent peritonitis	20 (4.2%)	16 (18.4%)	
Localized faecal peritonitis	3 (0.6%)	2 (2.3%)	
Diffuse faecal peritonitis	2 (0.4%)	1 (1.2%)	
	N = 474	N = 87	
Appendix aspect			<0.001
Erythematous	90 (18.99%)	3 (3.45%)	
Phlegmon	274 (57.81%)	19 (21.84%)	
Gangrene	87 (18.35%)	40 (45.98%)	
Perforation	23 (4.85%)	25 (28.74%)	
	N = 474	N = 87	
Peritoneal lavage			<0.001
Yes	332 (70.04%)	84 (96.55%)	
No	142 (29.96%)	3 (3.45%)	
	N = 474	N = 87	
Drainage			<0.001
Yes	223 (47.05%)	80 (91.95%)	
No	251 (52.95%)	7 (8.05%)	
	N = 474	N = 87	
(**c**)			
**Variable**	**DG 1–3**	**DG 4–5**	** *p* ** **-Value**
	**N = 474**	**N = 87**	
Length of stay	4.03 (±3.44)	6.14 (±3.07)	<0.001
	95% CI: [3.72; 4.34]	95% CI: [5.48; 6.79]	
	Range: (0.0; 21.0)	Range: (3.0; 21.0)	
	N = 474	N = 87	
Superficial SSI			0.397
Yes	2 (0.42%)	1 (1.15%)	
No	472 (99.58%)	86 (98.85%)	
	N = 474	N = 87	
Deep SSI			0.155
Yes	0 (0.0%)	1 (1.15%)	
No	474 (100.0%)	86 (98.85%)	
	N = 474	N = 87	
Organ/space SSI			<0.001
No	467 (98.52%)	74 (85.06%)	
Single abscess	6 (1.27%)	11 (12.64%)	
Multiple abscess	0 (0.0%)	2 (2.3%)	
Peritonitis	1 (0.21%)	0 (0.0%)	
	N = 474	N = 87	
New hospitalisation—30 days			0.099
Yes	9 (2.38%)	4 (6.35%)	
No	369 (97.62%)	59 (93.65%)	
	N = 378	N = 63	
New hospitalisation—60 days			>0.999
Yes	1 (0.26%)	0 (0.0%)	
No	377 (99.74%)	63 (100.0%)	
	N = 378	N = 63	

**Table 2 jpm-12-01904-t002:** Multivariable logistic regression model for technically challenging laparoscopic appendectomy, difficulty Grade 4–5. BMI, body mass index; ASA, American Society of Anaesthesiology classification; CCI, Charlson comorbidity index; AIR score, appendicitis inflammatory response score; CT, computed tomography.

Variable	Odds Ratio	*p*-Value
Constant	0.0105 [0.00289;0.0384]	<0.001
Age	1.02 [0.995;1.05]	0.108
ASA	0.653 [0.241;1.77]	0.401
CCI	1.21 [0.851;1.73]	0.285
AIR score	1.22 [1.06;1.4]	0.0043
Preoperative CT scan	2.38 [1.24;4.57]	0.009
BMI > 30 kg/m^2^	2.61 [1.29;5.27]	0.007

## Data Availability

The data presented in this study are available on request from the corresponding author. The data are not publicly available due to ethical issues.

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
