# Peer review of "Factors Influencing the Difficulty and Need for External Help during Laparoscopic Appendectomy: Analysis of 485 Procedures from the Resident-1 Multicentre Trial"

_jpm, 2022, doi:10.3390/jpm12111904_

Round 1

Reviewer 1 Report

This article identifies risk factors for complicated appendectomies requiring external help from the attending surgeon.

The introduction should focus on this background, and not get lost in topics without relevance to the article (non-operative management of acute appendicitis, open or laparoscopic appendectomy...).

Sentences such as:

- Recent studies also reported the feasibility of nonoperative management for uncomplicated appendicitis.

- Another paper explored the impact of previous abdominal surgery on the technical difficulty of LA showing no differences in terms of operative time, conversion rates, and postoperative outcomes. 

There is a very large methodological bias that needs to be addressed: appendectomies that required conversion to open surgery cannot be excluded from the study. This group of patients represents the most complex appendectomies, which could not be resolved laparoscopically and therefore required the assistance of the attending surgeon. The study should be redone to include this group of patients, otherwise we would be concluding unrealistic and unreliable data.

Another thing that needs to be clarified is the role of the "assistant" mentioned in the classification. Is this person another resident, a junior surgeon, a section chief?

The age range of the two groups includes 5 and 7 years respectively. The exclusion criteria indicate that only patients over 18 years of age were analysed. This incongruity makes it inadmissible for publication in a serious journal.

It is not understood why drainage is left in more than 40% of the appendectomies in the first group, when only 18% are gangrenous and 4% have peritonitis. Does this mean that drains are being left in patients with phlegmonous or erythematous appendicitis? Unacceptable in 2022.

In the first group there is one patient with 70 days of hospital admission according to the range shown in the table. However, no complications justifying more than 2 months of hospitalisation are indicated (only 6 single abscesses and one peritonitis are described).

It is not understood who operates the appendicitis in each group. It is assumed that in the first group they are operated on by the resident with more or less help from the assistant, while in group 2 the help of someone more experienced is required. How is it possible that only 34% of the appendicitis in the first group were operated on by a resident? If a surgeon operates them, who assists him? How is it possible that 51 residents from 21 hospitals participated in the study, and only 180 appendicitis were operated on among all these residents? Each resident operated on an average of 3-4 appendicitis in the 2 years of this study? How can this be? What kind of aberration is this? How can they even name the study "Resident-1 prospective multicenter trial"? Is this some kind of joke or irony? They must have given it that name because it is the residents who are in charge of collecting the data for the study, because it is obvious that they do not operate.

This manuscript requires profound methodological modifications that invalidate the results of the study. An extensive review of the entire study should be undertaken.

Author Response

Dear colleague,

thanks a lot for taking time to carefully review our work.

I appreciate the effort to make our study better.

You can find attached the reply to your points.

Thanks a lot

Bests

Stefano PB Cioffi

Reviewer 2 Report

The authors are presenting study named: “Factors influencing difficulty and need for external help during laparoscopic appendectomy: analysis of 485 procedures from the Resident-1 multicenter trial”.

First thing that caught my eye is the nonstandardized 1st page, with authors listed as shown, with ORCIDS.

Second thing is that in the footnotes a year 2021 is standing…

Nevertheless, authors made an attempt to do a good and interesting study.

these are my remarks:

1.       Abstract: please use number instead partitioned long word number- it is not reader friendly.

2.       Introduction: please add the reference number for the sentence: “Laparoscopic appendectomy is one of the most commonly performed procedures and is the standard of care for AA”.

3.       Introduction: please explain the gangrene word in this sentence: “Complicated Appendicitis (CA), defined as acute appendicitis with associated peritonitis, rupture, gangrene, or intra-abdominal abscess (IAA), accounts for 14%–55% of all cases of appendicitis”.
Gangrene of what? gangrenous appendicitis? – if so, if there is no perforation this is not a complicated appendicitis.

4.       Introduction: “To date, studies exploring preoperative factors associated with the technical difficulty of LA are lacking”. The sentence should be to our best knowledge, and somewhat lacking. Furthermore, I do not agree with what is said by this:

- Ussia A, Vaccari S, Gallo G, Grossi U, Ussia R, Sartarelli L, Minghetti M, Lauro A, Barbieri P, Di Saverio S, Cervellera M, Tonini V. Laparoscopic appendectomy as an index procedure for surgical trainees: clinical outcomes and learning curve. Updates Surg. 2021 Feb;73(1):187-195. doi: 10.1007/s13304-020-00950-z.

- Kim CW, Jeon SY, Paik B, Bong JW, Kim SH, Lee SH. Resident Learning Curve for Laparoscopic Appendectomy According to Seniority. Ann Coloproctol. 2020 Jul;36(3):163-171. doi: 10.3393/ac.2019.07.20

- Lin YY, Shabbir A, So JB. Laparoscopic appendectomy by residents: evaluating outcomes and learning curve. Surg Endosc. 2010 Jan;24(1):125-30. doi: 10.1007/s00464-009-0691-0. 

- Esparaz JR, Jeziorczak PM, Mowrer AR, Chakraborty SR, Nierstedt RT, Zumpf KB, Munaco AJ, Robertson DJ, Pearl RH, Aprahamian CJ. Adopting Single-Incision Laparoscopic Appendectomy in Children: Is It Safe During the Learning Curve? J Laparoendosc Adv Surg Tech A. 2019 Oct;29(10):1306-1310. doi: 10.1089/lap.2019.0112.

- Pang NQ, Chua HW, Kim G, Tan MY, Bin Abdul-Aziz MND, Xu RW, Chen E, Teo SC, Khoo NX, Lomanto D, Tai BC, So JB, Chong CS. Structured Training for Laparoscopic Appendectomy for Residents (STAR Trial)-A Randomized Pilot Study. J Surg Res. 2021 Dec;268:363-370. doi: 10.1016/j.jss.2021.06.073.

- and more…

5.       Introduction should be more detailed, more sparse, better written.

6.       Materials and methods: I encourage authors to write what is REsiDENT-1 trial multicentre registry, in details, and not just cite theirs other study from which they extracted data.

7.       Was this a prospective data collection or retrospective, what type of study is this?

8.       Question for authors: why were the children and elderly patients excluded? <18 and >80. What is the reason for this?

9.       Please add reference to Likert scale

10.   Materials and methods: please clearly state outcomes of the study!

11.   Materials and methods: I am confused why are the exclusion criteria as it stands!

in example:

-Operative conversion clearly means that resident or senior surgeon had some difficulty, therefore this way a data pool is not correct- confounding factor!

-Previous appendectomy??? – How can you perform an appendectomy if there way an appendectomy?

-Co existence of other infections? – this means that the operation was a bit harder? I don’t understand, you evaluate laparoscopic appendectomy level needed for help but you exclude all kinds of factors that happen in real life. In addition to this: previous appendicitis treated conservatively – why is this excluded???

This is a major flaw of the study on my account!

12.   Results: please make tables more reader friendly. Not to say that is absurd for the table to be across 4 pages. Divide it and make it more reader friendly.

13.   Results: there is a flowchart, figure 1 that clearly shows that 76 patients were excluded due to missing data. 44 patients lost at follow up. – there were no mention of this type of exclusion and inclusion at criteria in materials and methods.

14.   P values DO NOT need to be on 6 decimal digits.

The paper is very hard to follow, very reader unfriendly, with unclear aims and results. 

Author Response

(The authors gave the same response as above.)

Round 2

Reviewer 1 Report

The authors have corrected the main methodological errors in the manuscript. However, they should correct the style and minor typographical and grammatical errors.

Reviewer 2 Report

The authors did answer and correct all that was asked but, having the limited options regarding the basic setting of this study, primarily using Resident1 registry and all its flaws (incomplete data, multivariate exclusions, conversions) my opinion is that no serious conclusions can not be drawn in this kind of study setting.